# Host-pathogen interaction: *Enterobacter cloacae* exerts different adhesion and invasion capacities against different host cell types

Elisabet Frutos-Grilo[1¤], Vanessa Kreling[2], Andreas Hensel[2]*, Susana Campoy[1]

1 Departament de Genètica i de Microbiologia, Universitat Autònoma de Barcelona, Bellaterra, Spain,
2 Institute of Pharmaceutical Biology and Phytochemistry, University of Münster, Münster, Germany

¤ Current address: Centre de Regulació Genòmica, Parc de Recerca de Biomedicina de Barcelona, Barcelona, Spain
* ahensel@uni-muenster.de

**Data Availability Statement:** All relevant data are within the paper and its Supporting Information files.

## Abstract

New antibiotics are urgently needed due to the huge increase of multidrug-resistant bacteria. The underexplored gram-negative bacterium *Enterobacter cloacae* is known to cause severe urinary tract and lung infections (UTIs). The pathogenicity of *E. cloacae* in UTI has only been studied at the bioinformatic level, but until now not within systematic *in vitro* investigations. The present study assesses different human cell lines for monitoring the early steps of host-pathogen interaction regarding bacterial adhesion to and invasion into different host cells by flow cytometric adhesion assay, classical cell counting assay, gentamicin invasion assay, and confocal laser scanning microscopy. To our knowledge, this is the first report in which *E. cloacae* has been investigated for its interaction with human bladder, kidney, skin, and lung cell lines under *in vitro* conditions. Data indicate that *E. cloacae* exerts strong adhesion to urinary tract (bladder and kidney) and lung cells, a finding which correlates with the clinical relevance of the bacterium for induction of urinary tract and lung infections. Furthermore, *E. cloacae* ATCC 13047 barely adheres to skin cells (A-431) and shows no relevant interaction with intestinal cells (Caco-2, HT-29), even in the presence of mucin (HT29 MTX). In contrast, invasion assays and confocal laser scanning microscopy demonstrate that *E. cloacae* internalizes in all tested host cells, but to a different extent. Especially, bladder and kidney cells are being invaded to the highest extent. Defective mutants of *fimH* and *fimA* abolished the adhesion of *E. cloacae* to T24 cells, while *csgA* deletion had no influence on adhesion. These results indicate that *E. cloacae* has different pattern for adhesion and invasion depending on the target tissue, which again correlates with the clinical relevance of the pathogen. For detailed investigation of the early host-pathogen interaction T24 bladder cells comprise a suitable assay system for evaluation the bacterial adhesion and invasion.

**Funding:** SC: grants from the Fundació La Marató de TV3 (PI616163 TV3-201806-10) and Ministerio de Ciencia e Innovación (PID2020-117708GB-I00). EF-G: EMBO-award Short-Term Fellowship and finance its stay at University of Münster (no grant number available).

**Competing interests:** The authors have declared that no competing interests exist.

## Introduction

*Enterobacter* species, mostly *E. cloacae* and *E. aerogenes*, are responsible for causing many opportunistic nosocomial infections, and less commonly, community-acquired infections [1]. The clinical manifestations associated with *Enterobacter* spp. are diverse and often not distinguishable from those caused by other pathogens. Thus, *Enterobacter* spp. may cause urinary tract infections (UTIs), bacteraemia, lower respiratory tract infections or surgical site infections, and it also can colonize intravascular devices and among others [2].

In 1997, Sanders & Sanders started to highlight the incidence of *E. cloacae* in nosocomial infections and the appearance of multidrug-resistant strains [3]. Generally, these infections are of endogenous origin, occurring mostly in immunocompromised patients. However, the pathogen is naturally resistant to ampicillin, amoxicillin/clavulanic acid, cephamycin and the 1st and 2nd generation of cephalosporins owing to chromosomally encoded *ampC* β-lactamase [4]. Recently surveillance studies demonstrate also an increase of carbapenem-resistant *E. cloacae* isolates [5, 6]. Thus, multidrug-resistant *E. cloacae* are becoming an emerging threat to public health [7], representing a serious economic and public health issue for clinical healthcare [8, 9]. Despite being catalogued as a critical priority pathogen, the pathogenicity and virulence of this bacterium persist without being deeply studied, especially when looking at UTI [10, 11].

One of the first steps in host-pathogen interaction is the specific recognition of the host cell by the bacterium, followed by physical interaction of bacterial adhesins–mostly proteins, LPS, or glycosylated compounds from the outer membrane–with complementary binding partners from the host cell and subsequent invasion into the cell by membrane fusion or endosomal uptake [12]. The ability to adhere to host cell surfaces, such as epithelial tissues, is required for successful colonization and establishment of the infection. Therefore, it is a widespread attribute of gram-negative bacteria to express adherence factors responsible for recognizing and binding to specific receptors of the host cell, thus enabling the bacteria to invade the host cells and colonize the respective tissue [12]. For the majority of bacterial pathogens, one of the essential strategies to achieve and to initiate virulence and infection is the specific recognition and adhesion to epithelial tissues, which is prerequisite for the subsequent invasion into the host cell [13]. This host-pathogen interaction can be studied by use of *in vitro* models, using host cells and bacteria in a coincubation system, as in the case of the uropathogenic *Escherichia coli* [14]. Additionally, since several years, bacterial adhesion has been recognized as a valuable target for combating infections by use of the so-called antiadhesive drug compounds, inhibiting the early host-pathogen interaction [15–18].

Thus, the screening for optimized lead compounds comprises promising projects. On the other side, it has to be kept in mind that different bacteria use different adhesion strategies. In this context, antiadhesive compounds have been recognized to be developed specifically against a specific bacterium, and they cannot be developed as a general tool against a broad range of pathogens. Therefore, *in vitro* drug screening for anti-adhesive or anti-invasive compounds against *E. cloacae* needs the development and validation of the respective cell-based assay systems. Nonetheless, *in vitro* cell culture protocols, measuring *Enterobacter* adhesion or invasion abilities have been conducted only in a low variety of cell lines, such as Caco-2, HEp-2, and HeLa [13, 19], any of them related to urinary tract or pulmonary tissue, which are clearly targets of *Enterobacter* infections [2]. Therefore, the present work compares the interaction of *E. cloacae* with cell lines originating from the urinary tract with that of other human epithelial tissues. This aims to supply evidence of our consistent *in vitro* model to broaden the knowledge of *E. cloacae* in UTIs. Furthermore, the herein obtained data indicate wide specificity of *E. cloacae* against the different cell types, not only regarding the bacterial adhesion but

also concerning the internalization into the eukaryotic cells. Using these data sets, human T24 bladder and A-498 kidney cells are described for the first time as suitable *in vitro* models for adhesion and anti-adhesion studies of *E. cloacae*.

## Material and methods

### Bacterial strains, eukaryotic strains, and growth conditions

All strains used in this work are listed in Table 1. *E. cloacae* ATCC 13047 (NCBI: txid716541) was used as type strain [13, 20]. *E. cloacae* strains were grown for 16 h at 37˚C/10% $CO_2$ in Luria Bertuni (LB) 1.7% agar plates. When needed, antibiotics were added into the LB agar plates at suitable concentrations (e.g., chloramphenicol 34 µg/mL (Roche), kanamycin 50 µg/mL (Applichem) and gentamicin 20 µg/mL (Applichem).

**Table 1. Bacterial and eukaryotic cells and plasmids.**

| Strain or plasmid | Relevant characteristic(s) | Source or reference |
|---|---|---|
| **Bacterial strains** | | |
| DH5α | *E. coli supE4 ΔlacU169 (φ80 ΔlacZ ΔM15) hsdR17, recA1, endA1, gyrA96, thi-1, relA1* | Clontech |
| ATCC 13047 | *E. cloacae* subs. *cloacae* (Jordan) wild type strain, Amp^R | ATCC |
| UA1953 | *E. cloacae ΔfimA*, Amp^R, Gm^R | This work |
| UA1954 | *E. cloacae ΔfimH*, Amp^R, Gm^R | This work |
| UA1955 | *E. cloacae ΔcsgA*, Amp^R, Gm^R | This work |
| UA1956 | *E. cloacae ΔfimA* pUA1108(Kan)::*fimA*, Amp^R, Gm^R, Kan^R | This work |
| UA1957 | *E. cloacae ΔfimH* pUA1108(Kan)::*fimA*, Amp^R, Gm^R, Kan^R | This work |
| UA1958 | *E. cloacae ΔcsgA* pUA1108(Kan):: *csgA*, Amp^R, Gm^R, Kan^R | This work |
| **Eukaryotic strains** | | |
| ATCC T24 (HTB-4) | Epithelial human urinary bladder cells isolated from a transitional carcinoma | Generous gift of Prof. Straube |
| ATCC A-498 (HTB-44) | Epithelial human urinary kidney cells isolated from a carcinoma | Generous gift of Dr. M. Hilgruber |
| ATCC A-431 (CRL-1555) | Epithelial human epidermic skin cells isolated from an epidermoid carcinoma | Generous gift of Prof. J. Jose |
| ATCC A-549 (CCL-185) | Epithelial human lung cells isolated from a carcinoma | Generous gift of Prof. S. Ludwig |
| ATCC Caco-2 (HTB-37) | Epithelial human colon cells isolated from a colorectal adenocarcinoma. Knowing that *E. cloacae* is present in the gut microbiome and the fact that this cell line has been used in other works, colon tissue cells were selected | European Collection of Authenticated Cell Cultures |
| HT29 (HTB-38) | Epithelial human colon cells isolated from a colorectal adenocarcinoma. | Adenocarcinoma colorectal, ATCC |
| HT29-MTX-E12 | Mucus-secreting subclones derived from HT29 treated with methotrexate. | Generous gift from Prof. K. Langer Haga clic o pulse aquí para escribir texto. |
| **Plasmids** | | |
| pKOBEG | Vector containing the λ Red recombinase system, Cm^R, temperature sensitive | Generous gift of Prof. G. M. Ghigo Haga clic o pulse aquí para escribir texto. |
| pVRL1 | Vector carrying a gentamicin cassette, Gm^R | Generous gift of Prof. P. Visca Haga clic o pulse aquí para escribir texto. |
| pKD4 | Vector carrying FRT-Kan construction, Amp^R, Kan^R | [21] |
| pUA1108 | pGEX 4T-1 derivative plasmid carrying without the GST fusion tag, carrying only the Ptac IPTG-inducible promoter and the lacI^q gene; Amp^R | [22] |
| pUA1149 | pUA1108 derivative plasmid containing the native *E. cloacae fimA* gene under the control of the Ptac promoter, Kan^R [pUA1108(Kan)::*fimA*] | This work |
| pUA1150 | pUA1108 derivative plasmid containing the native *E. cloacae fimH* gene under the control of the Ptac promoter, Kan^R [pUA1108(Kan)::*fimH*] | This work |
| pUA1151 | pUA1108 derivative plasmid containing the native *E. cloacae csgA* gene under the control of the Ptac promoter, Kan^R [pUA1108(Kan)::*csgA*] | This work |

## Cell culture

All cell lines used in this work are described in Table 1. Different human cell lines were chosen for the adhesion assays, reflecting different organs, which can be targeted according to *E. cloacae* infections.

Cell lines used in this work were routinely cultured with DMEM 4.5 g/L glucose, with stable glutamine, without sodium pyruvate and 3.7 g/L $NaHCO_3$ media (PAN Biotech), supplemented with 1% (v/v) fetal calf serum (FCS) (Merck), and 0.5% (v/v) penicillin-streptomycin (Merck) at 37˚C and 5% $CO_2$ (for T24, A-431, A-498 and A-549) or 10% $CO_2$ (Caco-2, HT29, HT29 MTX). For experiments, passages #68–80 for T24, #32–54 for Caco-2, #35–46 for HT29, #42–53 for HT29-MTX, #22–33 for A-431, #48–59 for A-498, and #21–32 for A-549 were used. Prior to each *in vitro* experiment, cells were inspected under inverted light microscope to ensure confluence and the absence of any anomaly.

## Adhesion assay

The assay for quantifying bacterial adhesion to eucaryotic host cells was performed as described recently for uropathogenic *E. coli* and *H. pylori* [23, 24], with some modifications to adapt it to *E. cloacae* particularities.

For the assay, $5 \times 10^5$ eukaryotic cells/wells were seeded into 12-well plates and incubated at 37˚C ($CO_2$ content in the atmosphere according to the respective cell line) for 48 h in DMEM, supplemented with 10% FCS and 0.5% pen/strep, up to 90 to 100% of confluence. The medium was removed, cells were washed $1 \times$ with PBS and incubated together with 900 μL of DMEM/FCS. A bacterial cell suspension was prepared by diluting 16-hours LB plate grown *E. cloacae* in DMEM/FCS and an $OD_{640\,nm}$ of 4 (after appropriate dilution to enter a valid photometric range). In cases where flow cytometry was used for quantification, the bacteria were stained with fluorescein isothiocyanate (FITC, Sigma). The labelling was performed by resuspending the bacteria in sterile saline solution (NaCl 150 mM, $Na_2CO_3$ 100 mM, pH 8.0) to an $OD_{640nm}$ of 8 (after appropriate dilution to enter a valid photometric range), corresponding to $2 \times 10^9$ CFU/mL. 900 μL of this suspension were incubated for 60 min at 37˚C together with 100 μL of a FITC solution (10 mg/mL). FITC-labelled bacteria were washed $3 \times$ with PBS (8 g/L NaCl, 0.2 g/L KCl, 1.44 g $Na_2HPO_4$ and 0.1 g $KH_2PO_4$) and resuspended in DMEM/FCS, adjusting the solution to an $OD_{640nm}$ of 4. All further steps with FITC-labelled *E. cloacae* were performed under light protection. The bacterial concentration was confirmed by serial dilutions and plating on LB agar plates, thus calculating the colony forming units (CFU)/mL.

In all cases, bacteria were added into the wells to allow their contact with the eukaryotic cells for 1 h at 37˚C/5-10% $CO_2$. For these experiments, different bacteria/eukaryotic cell ratios (BCR) and $CO_2$ concentrations (from 5–10%) were used. Uninfected control groups were used by investigating host cell lines without addition of any bacteria. Following the incubation, non-adhered bacteria were removed by washing $3 \times$ with PBS and those remaining were quantified.

For flow cytometric quantification, cells were detached by addition of 500 μL of trypsin/ EDTA (Merck). Trypsinization was stopped after 5 min by addition of 500 μL of FCS. Cells were pelleted for 4 min at $350 \times$ g and resuspended in 500 μL of DMEM/FCS. Fluorescence of the cell suspension was measured immediately by flow cytometry (CytoFLEX, Beckman Coulter, λ = 488 nm / 540 nm). For each replicate, at least 10.000 counts per sample from viable cell gate replica were performed. The mean relative fluorescence intensity (median, M) of each sample population was determined. Thus, the increased value of the median represents a higher number of FITC-labelled bacteria attached to a host cell. The relative adhesion was calculated from the measured median of the infected cells compared to that of the control without bacteria.

For classical adhesion assays [19, 25], after washing steps, cells were recovered by disrupting the host cells by 5 min of incubation in the presence of Triton X-100 (0.1% in PBS) with shaking at 400 rpm and vortexing some seconds every minute. The membrane disruption was ensured by visualizing the samples under the light field inverted microscope. After homogenization, the lysates containing total cell-associated bacteria were serially diluted in PBS and plated onto LB agar plates to enumerate adherent bacteria. To determine the bacterial adhesion, the recovered *E. cloacae* were compared with the initially added bacteria, giving rise to the number of adhered bacteria.

Data results from at least three independent experiments (different cell line passages, and different bacterial passages and different day).

## Invasion assay

The ability of *E. cloacae* to invade into the cell was determined using a gentamicin-protective assay [26, 27]. In this experiment, $5 \times 10^5$ eukaryotic cells/well were seeded into 12-well plates and incubated at 37˚C/5-10% $CO_2$ for approximately 48 h in DMEM, supplemented with 10% FCS and 0.5% pen/strep until confluence of 90 to 100% was reached. The medium was removed, cells were washed $1 \times$ with PBS and incubated with 900 µL of DMEM, supplemented with 10% FCS. The bacterial suspension [$OD_{640\ nm}$ = 4 (after appropriate dilution to enter a valid photometric range).] had been prepared from diluting 16 h LB plate grown *E. cloacae* in DMEM/FCS. For all cell lines investigated, a BCR of 100:1 was used. Following a 3 h coincubation at 37˚C/5-10% $CO_2$, eukaryotic cells were washed $1 \times$ and incubated for 1 h at 37˚C/5-10% $CO_2$ together with a gentamicin solution (100 µg/mL, Applichem) in DMEM/FCS to remove free and adhered bacteria, thus conserving intracellular bacteria. Subsequently, the gentamicin-treated cells were incubated for 5 min with Triton X-100 (0.1% in PBS) by shaking at 400 rpm and vortexing some seconds every minute. Membrane disruption and cell lysis was controlled by visualizing the samples under the light field inverted microscope. The bacteria concentration of the so obtained suspension was determined by plating on LB agar plates. As control, cells incubated without bacteria (non-infected control) was included.

The activity of gentamicin in effective killing the *E. cloacae* under the described conditions has been validated in pre-experiments (S1 Fig in S1 Data). Further, it has been previously shown that gentamicin has no effect in the viability of human cells and thus has been used within invasion assays [28–32]. All experiments were performed in triplicate.

For the posterior data evaluation, the median from the determined CFU/mL, obtained from the cell lysate from three independent experiments (different cell passages, different bacterial passages and different experimental day) was calculated.

## Visualization of adhered and internalized *E. cloacae* in T24 cells by confocal laser scanning microscopy

For visualization of the adhesion and invasion of FITC-labelled *E. cloacae* by confocal laser scanning microscopy, a glass coverslip (Ø 13 mm, VWR) inside a 12-well plate was treated with 200 µL of a solution of collagen I from rat tail (ChemCruz) (4 mg/mL in acetic acid 0.02 mol/L) for 1 h under UV light exposure (approximately 254 nm). Subsequently, the coverslips were air-dried. Afterwards, $5 \times 10^5$ T24 cells/wells were seeded into the 12-well plates with the coverslips and incubated at 37˚C/5% $CO_2$ for 48 h until 90 to 100% confluency. Cells were washed $1 \times$ with PBS. After incubation with 1 mL of AlexaFluor™ 594 conjugated to wheat germ agglutinin (WGA, Invitrogen) (25 µg/200 µL in PBS, staining with 50 µL) and DAPI (Carl Roth (0.35 Mol/L in PBS, staining with 200 µL) for 20 min at 37˚C/5-10% $CO_2$ under

light protection, cells were washed 3 × with PBS and incubated with 1 mL of DMEM/FCS and the FITC-labelled *E. cloacae* at a BCR of 100:1.

For monitoring bacterial adhesion 1 h of incubation was used, for invasion assay a 3 h period plus a 1 h of incubation with gentamicin solution (100 μg/mL) was used. Subsequently, supernatants were removed, the cells were washed 3 × with PBS, fixated for 15 min by use of paraformaldehyde 4%, and washed 1 × with PBS and 1 × with water and mounted onto slides using Fluoromount-G™ (Invitrogen) Samples were visualized under the confocal microscope LSM800 Axio Observer Z1 (Carl Zeiss) using AF568 ($\lambda_{Detection}$ 575–700 nm), FITC ($\lambda_{Detection}$ 505–575 nm) and DAPI ($\lambda_{Detection}$ 400–505 nm) channels and z-stacking of 15 slices per image. Micrographs were treated with ImageJ software [33] and analyzed by the Cross Section Viewer plugin [34]. Non-infected controls were prepared in a similar way, but without addition of bacteria to the cells; as expected, no FITC fluorescence was observed.

For statistical evaluation, the experiments were independently repeated with at least 3 biological experiments at different days using different biological passages.

## Construction of *E. cloacae* knock-mutants and complemented strains

*E. cloacae* Δ*fimA*, Δ*fimH* and Δ*csgA* strains were constructed according to the λRed recombinase-based gene replacement method [21]. Gentamycin cassette were obtained by PCR from pVRL1 vector [35] using suitable oligonucleotides (S1 Table in S1 Data). The oligonucleotides incorporate the homology regions of the *E. cloacae* target gene (S1 Table in S1 Data). The PCR products were transformed into the corresponding *E. cloacae* cells containing the pKOBEG plasmid [36] (Table 1).

Plasmids harboring the corresponding tagged genes were constructed using the HiFi DNA assembly cloning kit (NEB) and the appropriate oligonucleotides (S1 Table in S1 Data). All PCR products were purified, cloned into the pUA1108 overexpression vector [22, 37] and transformed into *E. coli* DH5α (Table 1) [21]. Then, *bla* gene were disrupted using ScaI restriction enzyme (NEB) and kanamycin cassette from pKD4 vector were cloned in that space. In all cases, gene substitution and plasmid constructions were confirmed by PCR and sequencing (Macrogen, Madrid).

## Statistical analysis

Results from three independent assays performed in each described experiments were analyzed by one-way ANOVA and Dunnett's posttest (compare all pairs of columns) for statistical analysis by GraphPad Prism 7 (GraphPad Software, Inc.). In all cases, first column was taken as control to compare the other columns. $P < 0.05$ was determined as statistically significant (*) and $p < 0.01$ as highly significant (**).

## Results

### *E. cloacae* ATCC 13047 adheres efficiently to urinary tract and lung cell tissues, but not to intestinal and skin cells

To investigate the potential of *E. cloacae* ATCC 13047T24 to adhere to different human cell types, representing different organs, bladder cells (T24), kidney cells (A-498), intestinal cells (Caco-2), lung cells (A-549) and cells from skin epidermis (A-431) were incubated at BCR 100:1 with FITC-labeled bacteria for 60 min, followed by flow cytometric quantification of host cells with adhering bacteria [16, 23]. Alternatively, classical quantification of bacterial adhesion was performed by CFU counting of the bacterial load [19, 25]. Using this kind of protocol, fluorescence staining of the bacteria is not required. However, the here described use of flow cytometry provides additional information on the status of the eukaryotic cells, such as

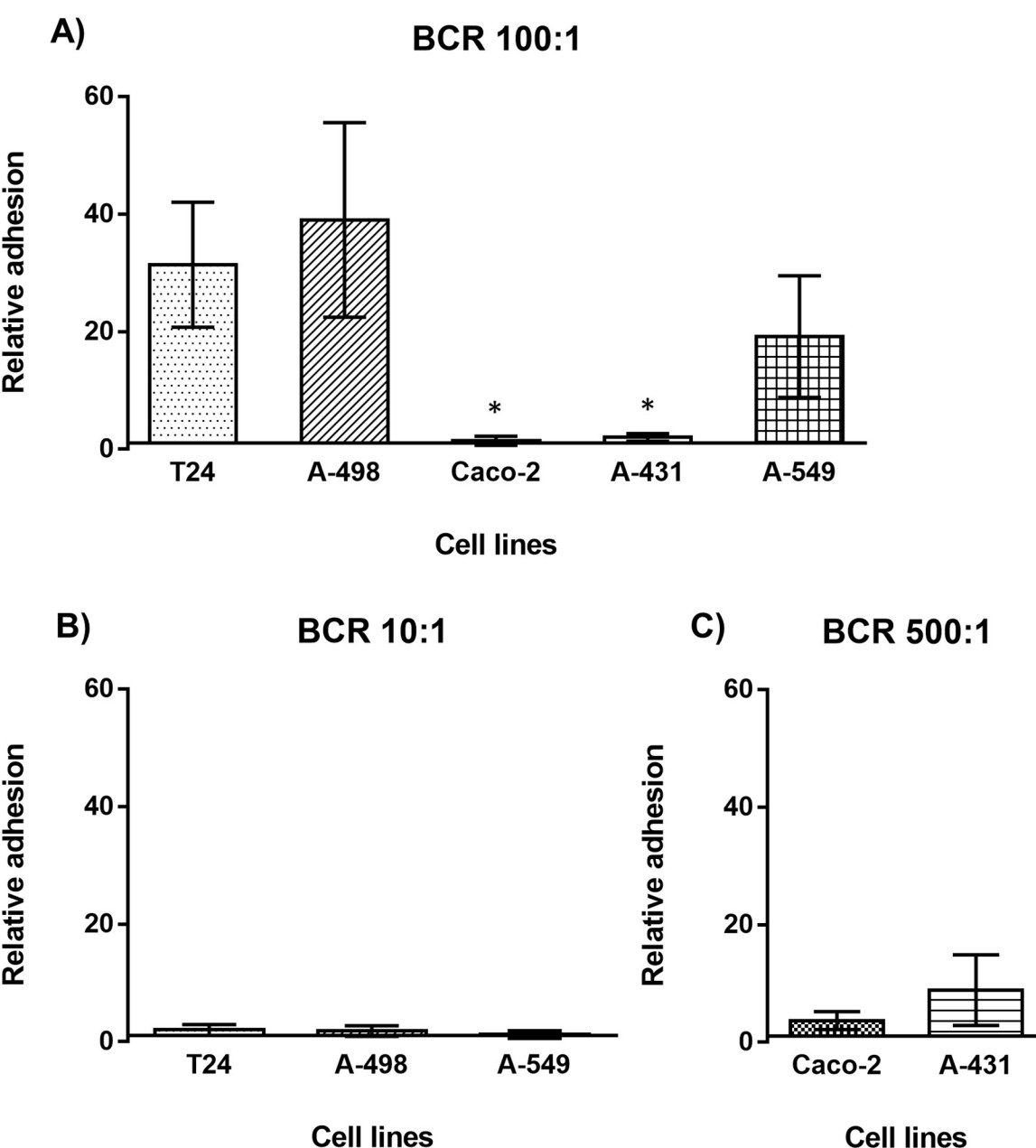

**Fig 1. Relative adhesion of FITC-labeled *E. cloacae* at different bacteria cell ratios (BCR) on cell lines from different tissues (T24 bladder cells, A-498 kidney cells, Caco-2 colon cells, A-431 epidermal cells and A-549 lung cells) as determined by flow cytometry.** **A**: BCR 100:1; **B**: BCR 10:1; **C**: 500:1. Relative data indicate the fluorescence related to uninfected cells. Values represent the mean ± SD from three independent assays and X-bar begins at Y = 1. *: $p < 0.05$, related to the relative adhesion of bacteria to T-24 cells.

cell death rate, discernment of cells with different number of adhered bacteria, or even cell differentiation determination [38].

Using this protocol, strong interaction of FITC-labelled *E. cloacae* with T-24 bladder cells, A-498 kidney cells, and A-549 lung cells was observed at BCR 100:1. A-431 and Caco-2 cells turned out to have no or only limited binding capacity against the bacterial treatment under these conditions and no adhesion or docking of the bacteria is observed (Fig 1A). Additionally, it got obvious that lower titers of *E. cloacae* load (BCR of 10:1) were not enough to initiate

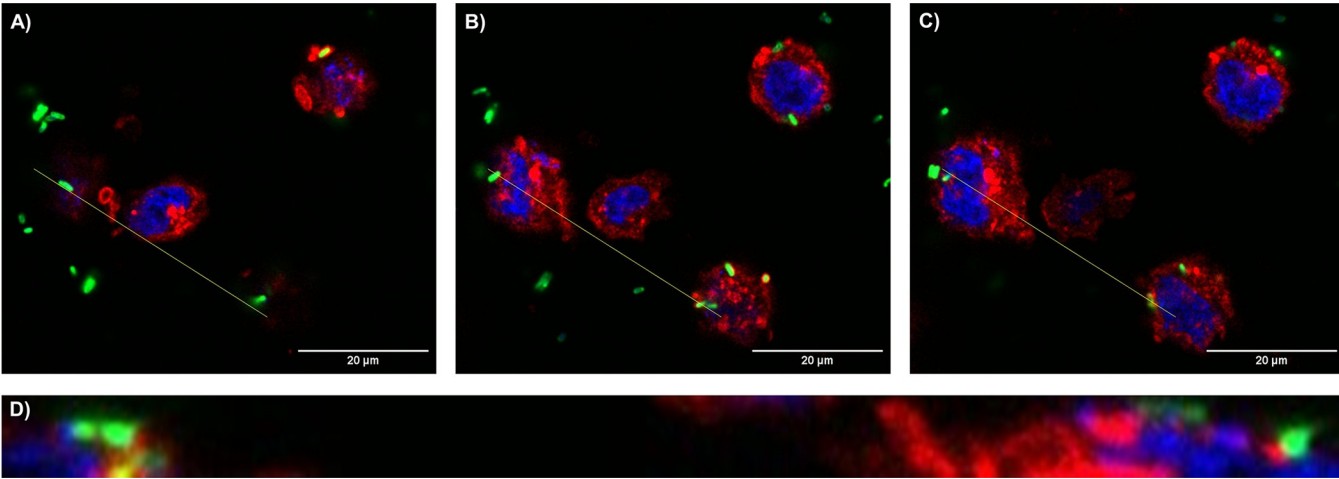

**Fig 2. Confocal laser scanning microscopy of T24 bladder cells infected with *E. cloacae* at BCR 100:1.** Different z-stack sections reveal representative (A) top, (B) middle, and (C) below sections of T24 cells. The cross section in *y* axis view from the yellow line in A, B, and C images is represented in (D). *E. cloacae* cells are displayed in green after staining with FITC, cell nuclei of T24 cells (blue) are stained with DAPI, and wheat germ agglutinin stain host cell membranes with AlexaFluor™ 594 (red).

relevant bacterial adhesion to T-24, A-498 and A-549 cells (Fig 1B). This could be because either a certain amount of bacteria is needed to infect the host cells or it is simply due to a very low fluorescence intensity as only a low amount of bacteria has been used in this protocol. Considering that *E. cloacae* displayed only low interaction with Caco-2 and A-431 cells at BCR 100:1, also BCR 500:1 was also employed for the adhesion assays with these cell lines (Fig 1C). However, this only results in a slight increase in fluorescence intensity, indicating that also increased bacteria level will not lead to more adhesion to Caco-2 and A-431 cells.

For validation of these results, it had to be ensured that the bacterial adhesion of *E. cloacae* to the host cells is not disturbed by exotoxin secretion of the bacteria. For that, filtered media of *E. cloacae* cultures were tested on T24, Caco-2, A-431, A-498 and A549 cells to assess potential cytotoxicity by MTT assay. Briefly, different cell free supernatants (CFS) were accomplished using different OD inoculations (0.2, 1.0 and 4.0) and times of growth (0, 30 min, 1 h, 3 h and 5 h) and incubated together with the cell lines over 48 h. As shown in S2 Fig in S1 Data no relevant influence of the test solutions on the cellular viability was observed. This indicates that the effects observed within the adhesion assay are due to the direct contact of *E. cloacae* with the host cells, as the filtered supernatants from *E. cloacae* cultures did not induce any significant cytotoxic effects against the different cell lines.

To confirm the results of the adhesion assay, the interaction of *E. cloacae* to the cell surface of the T24 cells at BCR 100:1 after 60 min of incubation was affirmed by confocal laser scanning microscopy (Fig 2). Location of FITC-labelled bacteria can clearly be assigned to occur on the outer side of the host cells. Internalization of the bacteria inside the cell was not observed within this protocol, indicating that bacterial invasion needs prolonged incubation times.

As mentioned above, *E. cloacae* is a commensal microbe from the gut, so we were surprised by the low bacterial adhesion observed to Caco-2 cells. Based on this finding other colonic cell lines, such as HT29 and HT29-MTX were integrated into this study, also to assess the possibility that the presence of mucin could prompt the adhesion. Mucin MUC-2 is produced only at very low levels low quantities by HT29 cells, while HT29-MTX cells are strong mucin producing cells after cell differentiation (approximately at day 21) [39, 40].

Every 2 to 3 days the media of Caco-2, HT29 and HT29-MTX cells were renewed to ensure optimal conditions and mucin droplet formation from HT29-MTX cells was confirmed by

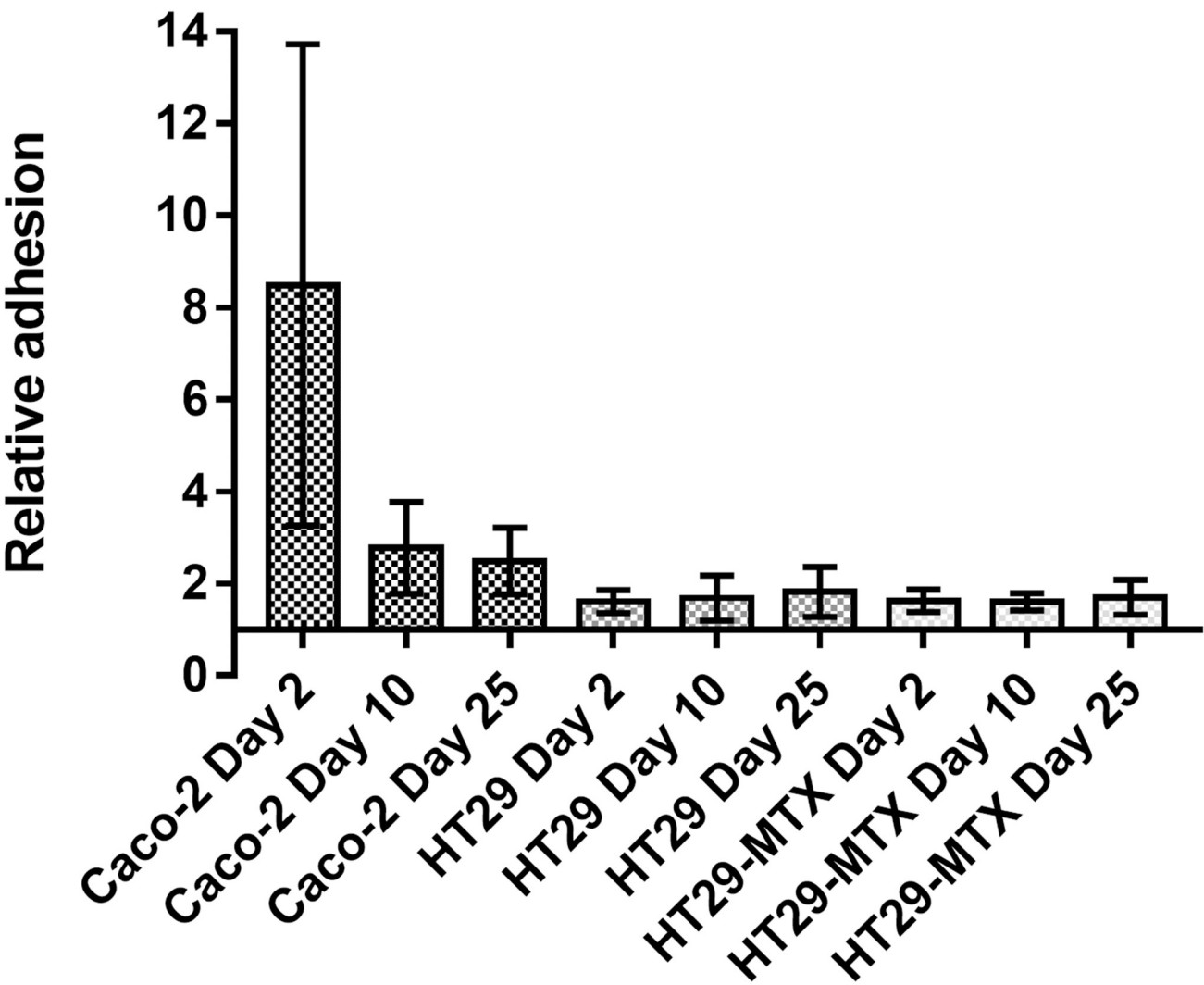

**Fig 3. Relative adhesion of FITC-labeled *E. cloacae* at BCR 500:1 to Caco-2, HT29 and HT29-MTX at different days of post-seeding (2, 10, and 25).** Relative data indicate the fluorescence related to uninfected cells and has been calculated from the median of the infected cells compared to that of the control without bacteria. Values represent the mean ± SD from three independent assays and X-bar begins at Y = 1.

microscopic observations. The *Enterobacter* adhesion ability to these cell lines was tested. Additionally, the study was performed at different cultivation times of the host cells to ensure that potential phenotypic changes during cultivation could be investigated for their putative influence on bacterial adhesion. As shown in Fig 3, *E. cloacae* exhibits lower adhesion onto HT29 and HT29-MTX than to Caco-2 cells. This corroborates the low interaction of *E. cloacae* to intestinal cells and demonstrates that mucin secretion seems to have no influence on the adhesion of *Enterobacter*.

### Cells from the urinary tract tissues, especially bladder cells, are more susceptible to *E. cloacae* internalization

It is not known whether *E. cloacae* is capable of invading urinary or respiratory cells. Therefore, and considering that adhesion is a necessary initial step to develop invasion, *in vitro* invasion assays were performed to investigate susceptibility of T24 bladder cells for *E. cloacae*

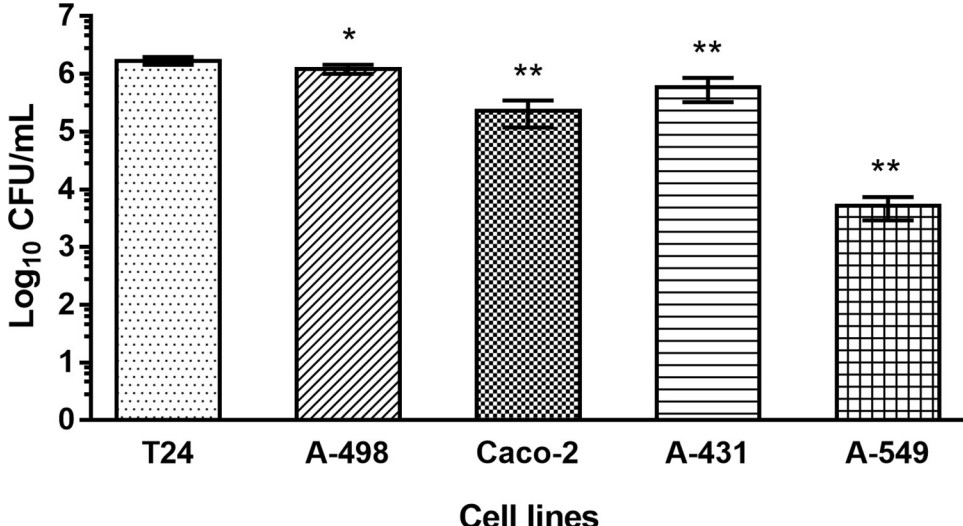

**Fig 4. *E. cloacae* load isolated from the different cell lines after invasion assay at BCR 100:1.** Data indicate the CFU/mL isolated from T24, Caco-2, A431, A498, or A-549 cells. Values represent the mean ± SD from three independent assays. One-way ANOVA and Dunnett's multiple comparison test (comparing all columns with T24) were used for statistical analysis. $P < 0.05$ was determined as statistically significant (\*) and $p < 0.01$ as highly significant (\*\*) compared to the bacterial invasion determined for T24 cells.

invasion. Furthermore, Caco-2, A-431, A-498 and A-549 cell were also analyzed. For invasion assay, incubation of the host cells with *E. cloacae* was performed for a contact time of 3 h, followed by the removal of non-invasive bacteria by gentamicin treatment. Internalized bacteria were released and seeded onto plates for quantification. Interestingly, *E. cloacae* was able to get internalized into all cell types, but to a different extent (Fig 4). T-24 bladder cells turned out to be most susceptible for invasion ($1.7 \times 10^6 \pm 2.5 \times 10^5$ CFU/mL), followed closely by A-498 kidney cells ($1.2 \times 10^6 \pm 2.2 \times 10^5$ CFU/mL), while A-431, Caco-2 and A-549 cells were invaded to a lower extent´ ($5.9 \times 10^5 \pm 2.6 \times 10^5$ CFU/mL, $2.3 \times 10^5 \pm 1.1 \times 10^5$ CFU/mL, $5.1 \times 10^3 \pm 2.2 \times 10^3$ CFU/mL, respectively).

Within the next step, confocal laser scanning microscopy was performed for specific monitoring of the internalization of the FITC-labelled *E. cloacae* into T24 cells, indicating strong accumulation of the bacteria inside the host cells (Fig 5).

## Type I pilus fimbriae, but not curli fimbriae, is required for the adhesion of *E. cloacae* onto bladder cells

For more detailed investigation of potential adhesive strategies of *E. cloacae* to interact with T24 bladder cells and to determine their virulence capability, different mutants were constructed and used for adhesion testing. For practical reasons quantification of bacterial adhesion was not performed by flowcytometric evaluation, but by use of classical adhesion assay, using cell counting. For preparation of mutants, the following genes were selected: *fimH* and *fimA*, which are regarded to be the major adhesin and the major subunit of type 1 pili respectively [41], and *csgA*, encoding for the major structural subunit of curli fimbriae [42].

The role of all these genes in the attachment to bladder cells has been clearly demonstrated for *E. coli* UPEC strains [43, 44]. Nevertheless, the association of either *fimH*, *fimA*, or *csgA* has never been determined in *E. cloacae*. Thus, to assay their implication on adhesion and to further validate T24 adhesion assay as a *E. cloacae* adhesion model, defective mutants for each gene were constructed by λRed system [19, 21]. Their adhesion activity to T24 cells was

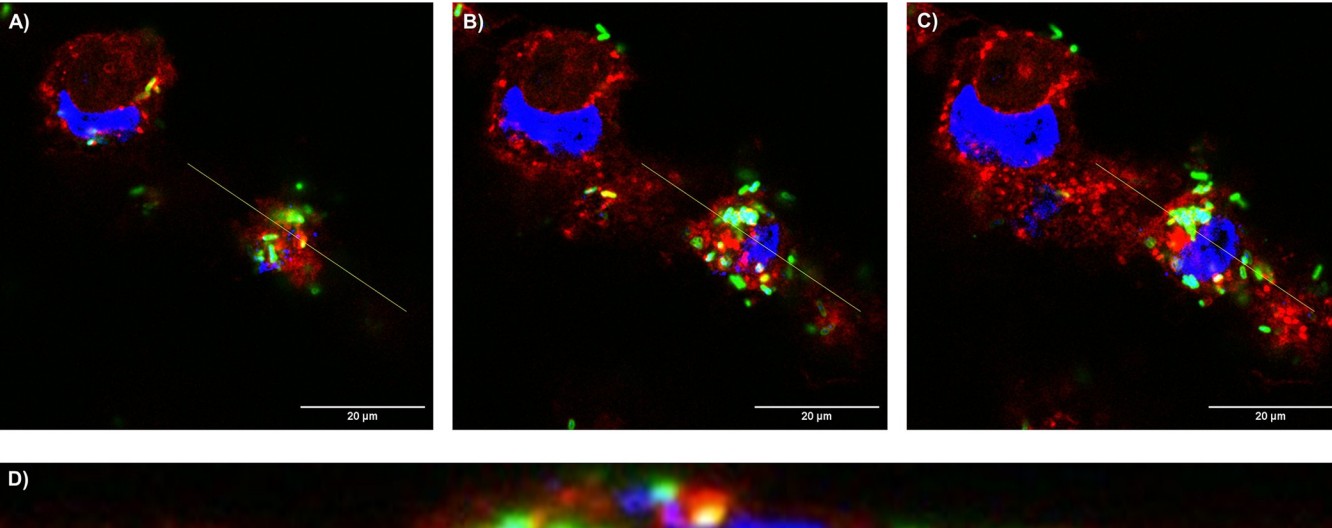

**Fig 5. Confocal laser scanning microscopy of T24 bladder cells infected with *E. cloacae* at BCR 100:1 for monitoring bacterial invasion into host cells.** Different z-stack sections reveal representative (A) top, (B) middle, and (C) below sections of T24 cells. The cross section in *y* axis view from the yellow line in A, B, and C images is represented in (D). *E. cloacae* cells are displayed in green after staining with FITC, cell nuclei of T24 cells (blue) are stained with DAPI, and wheat germ agglutinin stains host cell membranes with AlexaFluor™ 594 (red).

confirmed and it is shown in Fig 6. Interestingly, the deletion of both, *fimA* and *fimH* prompts out the absence of adhesion, as happens in *E. coli* UPEC, strains while no change with respect to that of the wild-type strain was observed in the absence of CsgA.

To further confirm these results, the Δ*fimA* and Δ*fimH* derivative strains were complemented using a plasmid carrying the corresponding deleted gene (Table 1, S4 in S1 Data). As shown in Fig 6, the presence of the plasmid carrying a copy of the deleted gene restores completely adhesion ability of the mutant strains, confirming unequivocally that FimA and FimH are needed for *Enterobacter* adhesion.

## Discussion

Bacteria must colonize host cells for the effective development of an infection. In this context, specific recognition and physical adhesion by protein-protein, protein-carbohydrate or carbohydrate-carbohydrate interaction is the first step prior to the invasion and/or secretion of toxins, thus it is a key event to be studied in bacterial pathogenesis [45]. For a better comparison and discussion, the results of the different adhesion and invasion assays obtained for the different cell lines have been summarized and relativized to T-24 cell as heat-map in Fig 7. Data from the present study provide information of adhesion and invasion properties of *E. cloacae* to different epithelial cell types, indicating strong preference of the bacterium towards cells of the urinary tract and bronchial tract (Fig 7). The described effects are assessed to be due to the direct contact of *E. cloacae* with the host cells, as the bacterium was shown not to induce significant cytotoxic effects against the different cell lines (S2 Fig in S1 Data).

Results visualized in Fig 7 clearly pointed out that the selection of the correct cell line is crucial for the development of cell-surface adhesion and invasion by *E. cloacae*. Overall, the obtained results indicate that the ability of *E. cloacae* to adhere to and invade into epithelial tissues is remarkably wide. For instance, T-24 and A-498 cells, both originating from the urinary tract, and A-549 from alveolar tissue, show the highest degree of interaction with *E. cloacae* at a BCR of 100:1. Additionally, a BCR of 10:1 was not enough to allow the adherence of higher

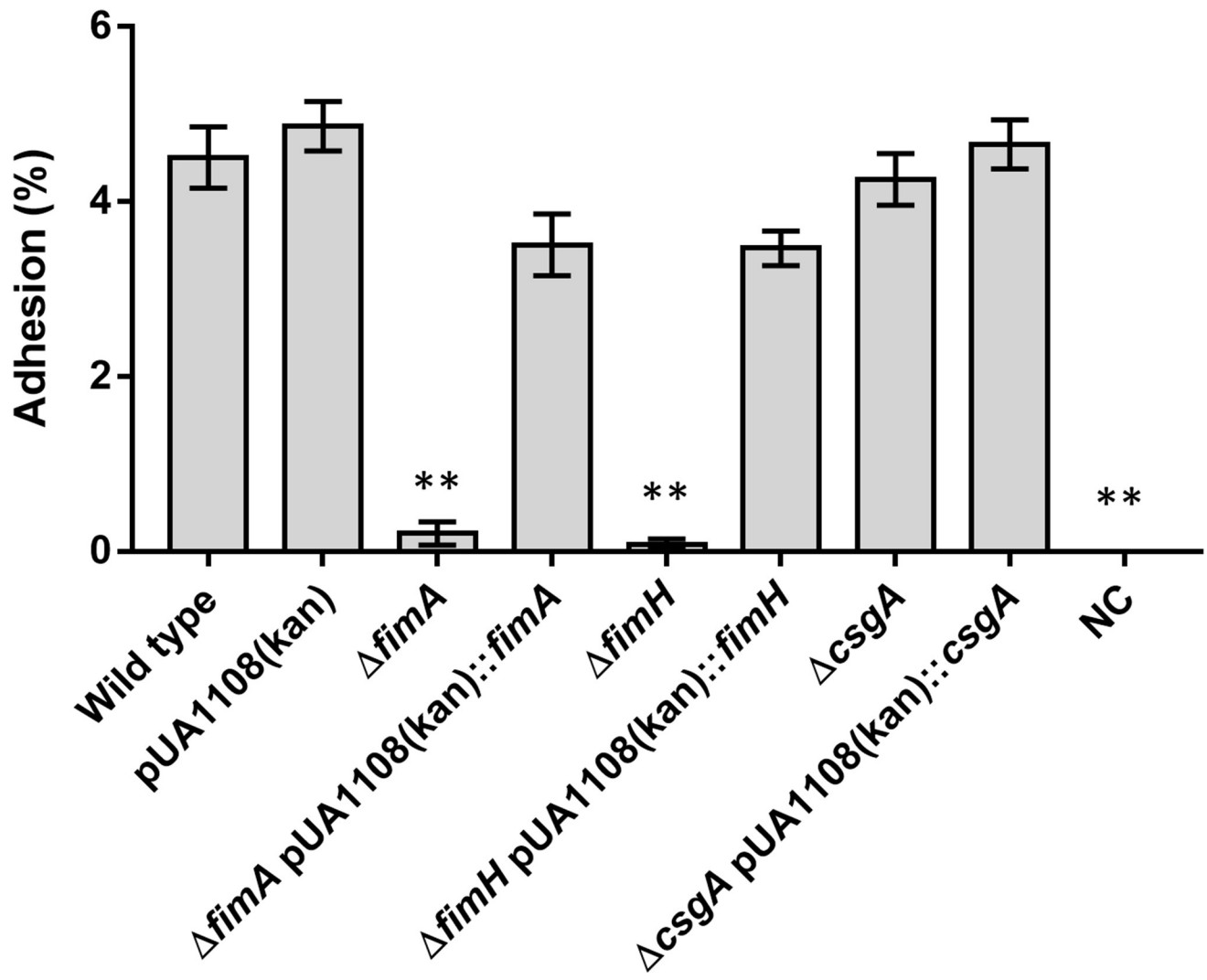

**Fig 6. Relative adhesion of *E. cloacae* to T-24 bladder cells (BCR 100:1, 60 min) observed in absence of *fimA*, *fimH*, or *csgA* genes, and complemented derivatives.** Data indicate the percentage of adhered bacteria in relation to the total bacterial count added to each sample (CFU/mL bacteria isolated after adhesion assay × 100 / CFU/mL bacteria added to each sample). NC: negative control (non-infected T-24 cells). Values represent the mean ± SD of three independent assays. **: $p < 0.01$, related to wild type bacteria.

amounts of bacteria to the host surface. The differences in the obtained results for these three tissues using the BCR of 10:1 and 100:1 were not proportional, demonstrating that a minimal load of inoculum might be required, where *quorum sensing* could be discussed for this phenomenon. UTIs represent the most common bacterial infections worldwide [46]. In fact, results obtained in this work point out that the main target for *E. cloacae* is indeed those tissues associated with the urinary tract (bladder and kidney). This is also in good correlation of epidemiological data sets for *Enterobacter*, responsible mainly for UTI and bronchial infections [47]. As intestinal cells are less infected by *E. cloacae* seems not surprising, as the gut is the normal habitat in humans for this organism. Also, infections of the skin by *E. cloacae* have been reported only very rarely [47].

| Cell line | BCR | Relative adhesion median | Relative invasion median |
|---|---|---|---|
| T24 | 10 | 5,1 | 130,8 |
| | 100 | 80,4 | 139,5 |
| Caco-2 | 100 | 3,7 | 19 |
| | 500 | 9,4 | 12,7 |
| A-431 | 100 | 5,2 | 48,3 |
| | 500 | 22,7 | 25,1 |
| A-498 | 10 | 4,6 | 154,6 |
| | 100 | 100 | 100 |
| A549 | 10 | 3,2 | 3,1 |
| | 100 | 49,1 | 0,4 |

**Fig 7. Relative adhesion and invasion results [%] of *E. cloacae* ATCC 13047 at BCR 100:1 against different cell lines, characterizing different tissue types.** Relative data are normalized against the interaction with T-24 bladder cells (= 100%). The degree of adhesion and invasion is colored by the heat map as displayed by the color bar legend. Identities closer to 0% are shown in red, and those higher than 100% in green.

However, due to the urinary tract anatomy and the acquisition of infection via the urethra, the risk of lower UTI progressing to pyelonephritis is low [46]. Considering this statement and the obtained data, the use of the herein T24 bladder cells described model to follow up studies on the different virulence factors of *E. cloacae* and its strategies for infecting cells might be interesting. In fact, and regarding the evaluation of the adhesion strategies using T24 model, our investigations of knock-out mutants strongly indicate that the absence of *fimA*, *fimH* genes significantly decreased the adhesion ability of *E. cloacae* to the host cells. Interestingly, the bacterial adhesion of these mutants was at a very low level, which might indicate that the products of *fimA* and *fimH* seem to be important adhesins of *E. cloacae* and that other adhesins are either not expressed or do not contribute to a higher extent to the specific interaction with the bladder cells. These *in vitro* finding should be investigated in further studies also within *in vivo* infections studies. As the fimbrial adhesins from UPEC are typical mannose-binding proteins, we hypothetise that the interaction of *E. cloacae* with T-24 bladder cells is due to a recognition of highly mannosylated uroplakin Ia and IIIa of the bladder cells, known to be the binding partner of the lectin-like domain of FimH from UPEC [48].

It must be noted that Pili type 1 (PT-I) are involved in binding to collagen type I and biofilm formation [49, 50], and, in contrast to PT-III, PT-I are present on *E. cloacae* ATCC 13047 proteome (S1 File in S1 Data). In that context, the deficiency of *fimA* or *fimH* genes, both needed for the correct assembly of PT-I [41], showed a strong reduction of the ability to adhere onto the T24 bladder cells. That result is not surprising because these genes are also crucial for other bacterial strains, but it is the first time observed in *E. cloacae*. Furthermore, the herein constructed Δ*fimA* and Δ*fimH* mutants are perfect controls for adhesion studies using T24 bladder cells as a cell line model.

On the other hand, curli fimbria is known to be associated to the adhesion related with human UPEC-induced cystitis, for instance over HTB-9 [42] and HTB-5 [43]. On the contrary, our results clearly provided evidence that they are not associated with *E. cloacae* adhesion to T24 since the Δ*csgA* mutant display the same adhesion phenotype than the wild-type strain, and as shown by qPCR the strain ATCC13047 expresses curli under the test condition (S5 Fig in S1 Data). These contrary results can be explained in cases where we understand the phenotypic differences between cell lines even originating from the same organ (e.g., from the

bladder epithelium), but from different human transitional carcinoma stages (e.g., grade I for T24, grade II for HTB-9 and grade IV for HTB-5) [51]. These differences can be correlated to different expression of bacteria-binding proteins, different TLRs or different glycosylation pattern of the cell matrix. Unfortunately, no further information on the proteome and glycome of the different cell types is available at the moment for a deeper discussion.

With respect to A-549 cell line-adenocarcinomic human alveolar basal epithelial cells, it displayed a highly sensitive ability to adhesion by *E. cloacae* (Fig 1). Alternatively, invasiveness is surprisingly insignificant (Fig 4), demonstrating once more a cell-dependent adhesion and/or invasion ability by *E. cloacae*. Pneumoniae caused by *E. cloacae* has been reported and discussed in detail [52, 53] and especially the high mortality rate due to manyfold and strong complications leads to hospitalization of most patients in intensive care units. It might be discussed for pneumonia, that *E. cloacae* may prefer to expand through the inner-alveolar surface, elevating the load of bacteria and precluding the correct $CO_2/O_2$ interchange within the lungs.

In contrast to T24, A498 and A-549 epithelial cells, the adhesion results onto A-431 and, specially, Caco-2 cells, displayed the lowest interaction values. The results of Caco-2 cells seem surprising due to the usage of these cell line in former adhesion studies with other *Enterobacter* species [54]. Also, other intestinal cells (HT29 from colorectal adenocarcinoma and the mucin-secreting cell line HT29-MTX) are not sensitive to adhesion of *E. cloacae*.

For that reason, we expected a higher degree of adhesion to intestinal cells Caco-2 cells, and the results were not improved using other mucin-producer cell lines such as HT29 and HT29-MTX, as shown in Fig 3. It has to be kept in mind that *E. cloacae* is part of the normal microbiota of the gut of 40 to 80% of the population and is also found in a high proportion of sewage samples, at concentrations up to $10^7$ organisms per gram [55, 56]. The essential part of the intestinal barrier function, the mucin, has been described to help the colonization of different strains such as *Lactobacillus rhamnosus*, *Streptococcus gallolyticus* and *Clostrydium butyricum* [57–59]. Nonetheless, the present results also demonstrated that MUC-2, the major gel-forming mucin present on colonic mucus [60], did not affect the adhesion values of *E. cloacae*. Pilus type 3 (PT-III) is required for binding to colonic mucus [58], which is not present in that strain. However, despite the low adhesion values (but existing) onto the colonic cells, *E. cloacae* is able to properly invade them although to a much lower extent compared to uroepithelial tested cells. Hence, its capacity to invade Caco-2 cells with a low previous adhesion may exhort a rapid and easy translocation of bacteria into these host cells. Nonetheless, further work is needed to understand the molecular mechanisms that allow this *Enterobacter* translocation inside the eukaryotic cells.

Overall, our results demonstrated that a high surface adhesion does not strictly means a good translocation into the colonized cell, and neither does a tissue with a high density of internalized bacteria means they were well attached to those eukaryotic cells. However, our results support strong evidence that T24 bladder cell line is a suitable model for adhesion and invasion studies for an improved understanding of the claimed pathogenicity of this bacterium in UTIs. In addition, the herein constructed *E. cloacae fimA* or *fimH* defective strains can be used as negative controls for adhesion assays. Further work is needed to increase the understanding of the pathogenicity of *E. cloacae* in more complex *in vitro* procedures, such as in organoids or organ-on-a-chips, which also give the advantage of methods without animal management.

## Supporting information

**S1 Data.**
(DOCX)

## Acknowledgments

We want to thank Gabriela Ortiz and Pau Conill for their technical support.

## Author Contributions

**Conceptualization:** Elisabet Frutos-Grilo, Andreas Hensel, Susana Campoy.

**Data curation:** Elisabet Frutos-Grilo, Andreas Hensel, Susana Campoy.

**Formal analysis:** Susana Campoy.

**Funding acquisition:** Susana Campoy.

**Investigation:** Elisabet Frutos-Grilo, Vanessa Kreling, Andreas Hensel, Susana Campoy.

**Methodology:** Elisabet Frutos-Grilo, Vanessa Kreling, Andreas Hensel.

**Project administration:** Andreas Hensel.

**Resources:** Andreas Hensel, Susana Campoy.

**Supervision:** Andreas Hensel, Susana Campoy.

**Validation:** Elisabet Frutos-Grilo.

**Visualization:** Elisabet Frutos-Grilo, Susana Campoy.

**Writing – original draft:** Elisabet Frutos-Grilo, Vanessa Kreling.

**Writing – review & editing:** Elisabet Frutos-Grilo, Andreas Hensel, Susana Campoy.

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
