## [Decision Letter · Decision Letter 0]

30 Mar 2023

PONE-D-23-05465Host-pathogen interaction: Enterobacter cloacae exerts different adhesion and invasion capacities against different host cell typesPLOS ONE

Dear Prof. Dr. Hensel,

Thank you for submitting your manuscript to PLOS ONE. After careful consideration, we feel that it has merit but does not fully meet PLOS ONE’s publication criteria as it currently stands. Therefore, we invite you to submit a revised version of the manuscript that addresses the points raised during the review process.

We look forward to receiving your revised manuscript.

Kind regards,

Kwame Kumi Asare, Ph.D

Academic Editor

PLOS ONE

Journal Requirements:

2. Thank you for submitting the above manuscript to PLOS ONE. During our internal evaluation of the manuscript, we found significant text overlap between your submission and previous work in the [introduction, conclusion, etc.].

Please revise the manuscript to rephrase the duplicated text, cite your sources, and provide details as to how the current manuscript advances on previous work. Please note that further consideration is dependent on the submission of a manuscript that addresses these concerns about the overlap in text with published work.

[If the overlap is with the authors’ own works: Moreover, upon submission, authors must confirm that the manuscript, or any related manuscript, is not currently under consideration or accepted elsewhere. If related work has been submitted to PLOS ONE or elsewhere, authors must include a copy with the submitted article. Reviewers will be asked to comment on the overlap between related submissions (http://journals.plos.org/plosone/s/submission-guidelines#loc-related-manuscripts).]

We will carefully review your manuscript upon resubmission and further consideration of the manuscript is dependent on the text overlap being addressed in full. Please ensure that your revision is thorough as failure to address the concerns to our satisfaction may result in your submission not being considered further

Reviewers' comments:

Reviewer's Responses to Questions

**Comments to the Author**

1. Is the manuscript technically sound, and do the data support the conclusions?

Reviewer #1: Yes

2. Has the statistical analysis been performed appropriately and rigorously? 

Reviewer #1: Yes

3. Have the authors made all data underlying the findings in their manuscript fully available?

Reviewer #1: Yes

4. Is the manuscript presented in an intelligible fashion and written in standard English?

Reviewer #1: Yes

5. Review Comments to the Author

Reviewer #1: The authors investigated the adhesion of Enterobacter cloacae isolate ATCC 13047 to different epithelical cell lines, incl. different urinary tract, intestinal tract, lung and skin cell lines using FITC labelled bacteria followed by FACS-based quantification of eukaryotic cell-associated fluorescence signals. They describe that the tested E. cloacae isolate adhered better to cell lines derived from the urinary tract and bronchial tract than to intestinal epithelial cells or a skin-derived cell line. This is interesting, because E. cloacae is known as an intestinal colonizer, but also as a pathogen causing urinary tract infection as well as respiratory tract infection. E. cloacae did not only adhere better to T-24 and A-498 cells, but also displayed a more pronounced invasive phenotype into these two urinary tract cell lines relative to cell lines from other parts of the body. Comparison of the adhesion of the wild type strain versus defined deletion mutants lacking fimH, fimA or csgA, the authors provide evidence that direct interaction between the bacteria and the cell line mediated by type 1 fimbriae, but not curli fimbriae is required for E. cloacae adherence.

This study is interesting in that there is no published data on the possible tissue tropism of E. cloacae or the importance of type 1 or curli fimbriae for the interaction between E. cloacae and host cells.

l. 245-259, Fig. 1 as well as l. 399-400: This reviewer does not think the wording is correct here. The disadvantage of the FACS-based method for quantifying bacterial adherence is that unfortunately no absolute numbers of adherent bacteria are obtained. This would have been more helpful here. This reviewer believes that the relative adherence determined with an MOI of 10:1 does fit the data obtained with an MOI of 100:1. Just because fewer bacteria were used does not mean they are better at binding. The relative adherence seems to be proportional to that obtained for the 100:1 MOI. The binding of E. cloacae to T24 and A-498 cells is thus significantly better than to Caco-2 and A-549 cells and about twice as good as to A-549 cells. The data would be even more convincing if the authors would, as a proof of principle, exemplarily quantify the number of adherent bacteria by CFU determination according to the quantification of the invasive bacteria. This reviewer is not convinced that a minimal load of inoculum or quorum sensing is required here.

l. 367-371, Fig. 6: The authors describe that subcloning of fimA, fimH and csgA in trans can restore the bacterial adherence phenotype to T-24 cells. The data presented looks nice, but it would be even nicer if the authors could provide additional evidence (SDS Page, ELISA, …) that indeed the subcloned gene results in expression of the corresponding gene product.

l. 423-430: The authors should phrase this sentence more carefully. According to the genome sequence, isolate ATCC 13047 has several fimbrial determinants. What do the authors know about the expression of different adhesins under the conditions that they have used? Can they prove that other fimbriae than type 1 fimbriae are not involved in bacterial adherence to T-24 or A-498 cells?

l. 432-434: Please phrase more carefully. Can the authors show that strain ATCC 13047 expresses curli under the test conditions?

The study would benefit from more careful visualization of adherent bacteria by electron microscopy or atomic force microscopy.

Another convincing proof that indeed adhesion mediated by type 1 fimbriae would be to demonstrate that bacterial adherence can be blocked by prior pre-incubation of the bacteria with e.g. methyl-α-D-mannopyranoside.

It would be interesting to see if the fimA, fimH and csgA mutants differ from the wild type strain in their ability to form biofilms.

6. PLOS authors have the option to publish the peer review history of their article (what does this mean?). If published, this will include your full peer review and any attached files.

Reviewer #1: No

---

## [Author Response · Author response to Decision Letter 0]

6 Jun 2023

Response to Reviewers

Dear Editor,

Ladies and Gentlemen,

Thank you for providing us with the opportunity to revise our manuscript based on the valuable feedback from the reviewer. We appreciate the time and effort that the reviewer has taken to carefully evaluate our work.

We have addressed all the concerns in the revised manuscript. Below this text, we provide a detailed response to each of the points raised, in bold. We hope that these revisions meet with the satisfaction of the reviewers and the editorial board. Once again, we thank the reviewers and the editor for their valuable feedback and guidance, which have helped to improve the quality of our work.

Sincerely,

Andreas Hnsel in behalf of all authors

We have changed the file naming and other issues in order to ensure the PLOS ONE’s style requirements.

2. Thank you for submitting the above manuscript to PLOS ONE. During our internal evaluation of the manuscript, we found significant text overlap between your submission and previous work in the [introduction, conclusion, etc.].

Please revise the manuscript to rephrase the duplicated text, cite your sources, and provide details as to how the current manuscript advances on previous work. Please note that further consideration is dependent on the submission of a manuscript that addresses these concerns about the overlap in text with published work.

[If the overlap is with the authors’ own works: Moreover, upon submission, authors must confirm that the manuscript, or any related manuscript, is not currently under consideration or accepted elsewhere. If related work has been submitted to PLOS ONE or elsewhere, authors must include a copy with the submitted article. Reviewers will be asked to comment on the overlap between related submissions (http://journals.plos.org/plosone/s/submission-guidelines#loc-related-manuscripts).]

We will carefully review your manuscript upon resubmission and further consideration of the manuscript is dependent on the text overlap being addressed in full. Please ensure that your revision is thorough as failure to address the concerns to our satisfaction may result in your submission not being considered further.

All text has been carefully read to identify duplications using plagiarism checker servers (duplichecker.com and plagiarismdetector.net), trying to minimize the overlapping text with other works despite being referenced. We have realized an unreferenced and duplicated sentence, hence we have rephased it and referenced (LINE 70-72). However, we have not found any other sentence suspected to be duplicated. We would appreciate if you detect any other concerning sentence, let us know.

The included “Data not shown” has been eliminated from the manuscript but we described the results (LINES 211 AND 297) which reads now as follows: Non-infected controls were prepared in a similar way, but without addition of bacteria to the cells; as expected, no FITC fluorescence was observed.

Reviewers’ comments:

I. 245-259, Fig. 1 as well as l. 399-400: This reviewer does not think the wording is correct here. The disadvantage of the FACS-based method for quantifying bacterial adherence is that unfortunately no absolute numbers of adherent bacteria are obtained. This would have been more helpful here. This reviewer believes that the relative adherence determined with an MOI of 10:1 does fit the data obtained with an MOI of 100:1. Just because fewer bacteria were used does not mean they are better at binding. The relative adherence seems to be proportional to that obtained for the 100:1 MOI. The binding of E. cloacae to T24 and A-498 cells is thus significantly better than to Caco-2 and A-549 cells and about twice as good as to A-549 cells. The data would be even more convincing if the authors would, as a proof of principle, exemplarily quantify the number of adherent bacteria by CFU determination according to the quantification of the invasive bacteria.

This is a very interesting issue, thanks for this comment. The experiments indicated different sensitivity of the different cell lines. While T24, A-498 and A-549 cells were strongly infected by the bacteria at BCR 100:1 nearly no interaction has been shown for Caco-2 and A-431 cells (Fig. 1A). We have increased the BCR to 500:1 for Caco-2 and A-431 and again we do not find relevant fluorescence intensity (Fig. 1C). From these data we can clearly conclude that these cell lines are much less sensitive to E. cloacae compared to the bladder, kidney and lung cells. For the sensitive cell lines we used also a lower BCR of 10:1 but in this experimental set up we detect only very low intensity (Fig. 1B). This might have two reasons: Either a certain bacteria density if needed for infection or the fluorescence intensity is very low, as only 10% of the labelled bacteria load has been used in this experiment. We have inserted these both points into the text – our impression is that we are working in this protocol at the analytical limit of qunatitatioin in the FACS assay. Alternatively CFU counting could be performed, which normally works very well, but at this low BCR standard deviations are so high that a clear and statistical evaluation cannot be done. Concenring the advantages and disadvantages of the FACS assay raised by the referee, we have the feeling that flow cytometric data are getting pretty good results for adhesion assays, as this protocol has been established for quantitative evaluation of new adhesion and invasion blockers since some years, leading to less standard deviations as we normally see in CFU counting. 

The respective section in the MS reads now as follows: “….Using this protocol, strong interaction of FITC-labelled E. cloacae with T-24 bladder cells, A-498 kidney cells, and A-549 lung cells was observed at BCR 100:1. A-431 and Caco-2 cells turned out to be not sensitive against the bacterial treatment under these conditions (Figure 1A). Additionally, it got obvious that lower titers of E. cloacae load (BCR of 10:1) were not enough to initiate relevant bacterial adhesion to T-24, A-498 and A-549 cells, which (Figure 1B). This could be because either a certain amount of bacteria is needed to infect the host cells or it is simply due to a very low fluorescence intensity as only a low amount of bacteria has been used in this protocol. Considering that E. cloacae displayed only low interaction with Caco-2 and A-431 cells at BCR 100:1, also BCR 500:1 was also employed for the adhesion assays with these cell lines (Figure 1C). However, this only results in a slight increase in fluorescence intensity, indicating that also increased bacteria level will not leed to more adhesion to Caco-2 and A-431 cells.” 

This reviewer is not convinced that a minimal load of inoculum or quorum sensing is required here.

We are totally agree with the reviewer’s opinion and we have removed the sentence “The differences in the obtained results for these three tissues using the BCR of 10:1 and 100:1 were not proportional, demonstrating that a minimal load of inoculum is required, where quorum sensing could be discussed for this phenomenon.” in order to avoid to talk about non demonstrated facts. 

II. 367-371, Fig. 6: The authors describe that subcloning of fimA, fimH and csgA in trans can restore the bacterial adherence phenotype to T-24 cells. The data presented looks nice, but it would be even nicer if the authors could provide additional evidence (SDS Page, ELISA, …) that indeed the subcloned gene results in expression of the corresponding gene product.

We provide RT-PCR assays showing the expression of the fimA, fimH, and csgA genes. As is shown in the new Suppl. File S4, in all mutant strains, and as expected, no RNA of the deleted gene (csgA, fimA, or fimH) is detected. However, the expression of these genes is only detected when the pUA1108 (kan) plasmid carrying the corresponding wild-type gene is incorporated into the mutant strain. These results, together with those in Figure 6 of the manuscript unequivocally indicate that the presence of the corresponding plasmid restores the expression of each deleted gene. 

III. 423-430: The authors should phrase this sentence more carefully. According to the genome sequence, isolate ATCC 13047 has several fimbrial determinants. What do the authors know about the expression of different adhesins under the conditions that they have used? Can they prove that other fimbriae than type 1 fimbriae are not involved in bacterial adherence to T-24 or A-498 cells?

We don’t know about the expression of other adhesins present in the strain, for this reason we recommend further studies in order to investigate it. We rephrase the sentence: “As the fimbrial adhesins from UPEC are typical mannose-binding proteins, we hypothetise that the interaction of E. cloacae with T-24 bladder cells is due to a recognition of highly mannosylated uroplakin Ia and IIIa of the bladder cells, known to be the binding partner of the lectin-like domain of FimH from UPEC [54].”

Instead of:

“As the fimbrial adhesins from UPEC are typical mannose-binding proteins, we assume that the interaction of E. cloacae with T-24 bladder cells is due to a recognition of highly mannosylated uroplakin Ia and IIIa of the bladder cells, known to be the binding partner of the lectin-like domain of FimH from UPEC [54].”

IV. 432-434: Please phrase more carefully. Can the authors show that strain ATCC 13047 expresses curli under the test conditions? 

In order to demonstrate that E. cloacae ATCC13047 expresses curli under the test conditions, we provide a RT-qPCR results in S5 Figure of the Suppl. Data.

The study would benefit from more careful visualization of adherent bacteria by electron microscopy or atomic force microscopy. 

We concur with the reviewer's viewpoint and the study would benefit from more careful visualization of adherent bacteria by electron microscopy or atomic force microscopy. However, bacterial adhesion and invasion has been proven using confocal images, as we do, in different publications (1 doi.org/10.1016/S0076-6879(95)53016-9; 2 10.1038/s41598-020-63714-0; 3 10.1128/AEM.03323-12). Moreover, with the confocal images we have a 3D representation of the adhesion and invasion events (as presented in the Figures 2 and 5 of the present MS). 

Another convincing proof that indeed adhesion mediated by type 1 fimbriae would be to demonstrate that bacterial adherence can be blocked by prior pre-incubation of the bacteria with e.g. methyl-α-D-mannopyranoside.

This is of course a good positive control, which definitely will be integrated into future investigations.

It would be interesting to see if the fimA, fimH and csgA mutants differ from the wild type strain in their ability to form biofilms.

We agree with the reviewer interests and we add biofilm results of the ability of fimA, fimH and csgA mutants to form biofilm. The image has been added to the Supplementary Data S3) into the manuscript.

---

## [Decision Letter · Decision Letter 1]

12 Jul 2023

PONE-D-23-05465R1Host-pathogen interaction: Enterobacter cloacae exerts different adhesion and invasion capacities against different host cell typesPLOS ONE

Dear Dr. Hensel,

Thank you for submitting your manuscript to PLOS ONE. After careful consideration, we feel that it has merit but does not fully meet PLOS ONE’s publication criteria as it currently stands. Therefore, we invite you to submit a revised version of the manuscript that addresses the points raised during the review process.

We look forward to receiving your revised manuscript.

Kind regards,

Kwame Kumi Asare, Ph.D

Academic Editor

PLOS ONE

Journal Requirements:

Reviewers' comments:

Reviewer's Responses to Questions

**Comments to the Author**

1. If the authors have adequately addressed your comments raised in a previous round of review and you feel that this manuscript is now acceptable for publication, you may indicate that here to bypass the “Comments to the Author” section, enter your conflict of interest statement in the “Confidential to Editor” section, and submit your "Accept" recommendation.

Reviewer #2: (No Response)

Reviewer #3: All comments have been addressed

2. Is the manuscript technically sound, and do the data support the conclusions?

Reviewer #2: Yes

Reviewer #3: Yes

3. Has the statistical analysis been performed appropriately and rigorously? 

Reviewer #2: Yes

Reviewer #3: I Don't Know

4. Have the authors made all data underlying the findings in their manuscript fully available?

Reviewer #2: Yes

Reviewer #3: Yes

5. Is the manuscript presented in an intelligible fashion and written in standard English?

Reviewer #2: Yes

Reviewer #3: Yes

6. Review Comments to the Author

Reviewer #2: The article entitled ‘Host-pathogen interaction…..’ by Hensel et al. describes mechanism of pathogenicity of E. cloacae using different cell lines. Despite work on several pathogens, the pathogenicity of E. cloacae, a member of ESKAPE organisms, is still under-explored. This work investigates the adhesion and invasion of E. cloacae in cell lines which is important for understanding the pathogenicity of nosocomial pathogens. The work is well designed and executed. The authors have also incorporated the suggestions of other reviewers. However, I have few minor concerns which can be corrected or addressed.

Abstract

Line 35-36: Correct ‘extend’ to extent. Do the same across the text wherever it applies.

Line 126, 169: O.D of 4 or 0.4. Please confirm. 4 O.D seems too high.

Line 128: is it 8 or 0.8 OD?

Line 269: How do you rule out the possibility of T6SS-mediated cytotoxicity as you have used CFS? E. cloacae is known to possess T6SS, which causes contact-dependent cytotoxicity.

Fig 3: What is on Y axis

Line 430: Italicize species name.

Reviewer #3: I am seeing this manuscript for the first time (after it has already been revised, obviously). Overall, this is robust work and the previous referee comments have been addressed satisfactorily. I have some smaller comments to manuscript organisation and to phrasing/language:

1) Manuscript Organisation: supplementary material. I find it porblematic to "hide" normal figures in the supplementary material in an online journal that has no restriction on the number of figures for the main text. I feel that in PLOS ONE and comparable journals, supplementary material should be restricted to large datasets, movies, etc that cannot be displayed in a PDF properly. All other data can only fall into two categories: (A) important - then show it in the main manuscript and reference/discuss it. Or (B) not important: then do not show it and do not talk about it in the manuscript.

2) phrasing: I ma not very happy with the wording related to the binding/interaction data between cells and bacteria. It feels a bit "off" to me to say "A-431 and Caco-2 cells turned out to be not sensitive against the bacterial treatment" - what do you mean by "sensitive"? What you are observing is bacteria binding to cells, not cells 'reacting' to the bacteria, correct? I suggest to find a better way to describe the physical interaction (or non-interaction) that is observed here. This might apply to other sections of the text too.

7. PLOS authors have the option to publish the peer review history of their article (what does this mean?). If published, this will include your full peer review and any attached files.

Reviewer #2: **Yes: **Prabhat Nath Jha

Reviewer #3: No

---

## [Author Response · Author response to Decision Letter 1]

14 Jul 2023

Referees comments and action of the authors 

Comment of reviewer Action of authors

Reviewer #2: The article entitled ‘Host-pathogen interaction…..’ by Hensel et al. describes mechanism of pathogenicity of E. cloacae using different cell lines. Despite work on several pathogens, the pathogenicity of E. cloacae, a member of ESKAPE organisms, is still under-explored. This work investigates the adhesion and invasion of E. cloacae in cell lines which is important for understanding the pathogenicity of nosocomial pathogens. The work is well designed and executed. The authors have also incorporated the suggestions of other reviewers. However, I have few minor concerns which can be corrected or addressed. ´

Thanks for the positive evaluation!

Abstract

Line 35-36: Correct ‘extend’ to extent. Do the same across the text wherever it applies. 

Has been changed as requested (also over the MS, 5 �)

Line 126, 169: O.D of 4 or 0.4. Please confirm. 4 O.D seems too high.

 Line 128: is it 8 or 0.8 OD? The OD values of the cultures have been determined after appropriate dilution to enter a valid photometric range, but a given then for the undiluted text sample.

Line 269: How do you rule out the possibility of T6SS-mediated cytotoxicity as you have used CFS? E. cloacae is known to possess T6SS, which causes contact-dependent cytotoxicity. Thanks for this advice. In this experiment we have shown that the bacterial adhesion of E. cloacae to the different host cells is not disturbed by soluble exotoxin secretion by using cell free and filtered culture supernatants of the bacteria at different ODs. This experiments did not show any influence on the different host cells, indicating that no soluble bacterial toxins disturb the assay. Unfortunately we wrote in the old version the phrase “… to show that the adhesion of E. cloacae to the host cells is not disturbed by contact associated cytotoxicitry or by exotoxin secretion”. The reviewer is absolutely right, T6SS would be a problem in the cell systems, but as we used in this experiment soluble cell free systems this secretion system is not part of the test solutions. We have changed the respective paragraph and deleted “by contact-associated cytotoxicity”. 

Fig 3: What is on Y axis Y axis described the relative adhesion of FITC-labeled bacteria at BCR 500:1 to different host cells at different days. The Relative data indicate the fluorescence related to uninfected cells, which is stated in the legend. 

As mentioned in the corresponding Material and Methods “The relative adhesion was calculated from the measured median of the infected cells compared to that of the control without bacteria“, we have divided the given fluorescence of the infected cells, by that from the non infected control. So we have X times more fluorescence when we are adding the bacteria because it is adhered onto the epithelial cells. 

We have added this to the legend of Fig. 3, which now reads as follows:

“Fig 3. Relative adhesion of FITC-labeled E. cloacae at BCR 500:1 to Caco-2, HT29 and HT29-MTX at different days of post-seeding (2, 10, and 25). Relative data indicate the fluorescence related to uninfected cells and has been calculated from the median of the infected cells compared to that of the control without bacteria. Values represent the mean ± SD from three independent assays and X-bar begins at Y = 1.”

Line 430: Italicize species name. Has been changed as requested

Reviewer #3: I am seeing this manuscript for the first time (after it has already been revised, obviously). Overall, this is robust work and the previous referee comments have been addressed satisfactorily. I have some smaller comments to manuscript organisation and to phrasing/language: 

Thanks for the positive comments!

1) Manuscript Organisation: supplementary material. I find it porblematic to "hide" normal figures in the supplementary material in an online journal that has no restriction on the number of figures for the main text. I feel that in PLOS ONE and comparable journals, supplementary material should be restricted to large datasets, movies, etc that cannot be displayed in a PDF properly. All other data can only fall into two categories: (A) important - then show it in the main manuscript and reference/discuss it. Or (B) not important: then do not show it and do not talk about it in the manuscript. 

This is a difficult question which should be decided by the editor or by the journals management. From our point of view, the Suppl Material provides access of data to the reader of the MS, which is not absolutely essential to know, but prove and validate experimental data which again validate the main experiments and results (e.g. pre-experiments, validation studies, dose finding). Or they simply get access to some very special features of the materials used (in our cases detailed description of the E. cloacae mutants) or similar data, which are essential for good science but only interesting for a reader f he really wants to go in the lab experiments or wants to repeat experiments in his own lab. From our point of view the Suppl. Data provide a nice help for interesting scientists in the daily lab routine. 

Of course we easily can insert all our Suppl Data into the main MS, but we have the feeling that then the text could be overloaded and the main messages might be difficult to access. 

If the editor wants us to do this, we are happy to revise, but according to our feeling and experience in publishing > 300 papers it could be clearer to have a separation into the main data set and a Suppl. Data File. However, we are very open to potential improvements. 

2) phrasing: I ma not very happy with the wording related to the binding/interaction data between cells and bacteria. It feels a bit "off" to me to say "A-431 and Caco-2 cells turned out to be not sensitive against the bacterial treatment" - what do you mean by "sensitive"? What you are observing is bacteria binding to cells, not cells 'reacting' to the bacteria, correct? I suggest to find a better way to describe the physical interaction (or non-interaction) that is observed here. This might apply to other sections of the text too. Thanks for this advice, this is true.

We have changed the text accordingly, which reads now as follows:

“….Using this protocol, strong interaction of FITC-labelled E. cloacae with T-24 bladder cells, A-498 kidney cells, and A-549 lung cells was observed at BCR 100:1. A-431 and Caco-2 cells turned out to have no or only limited binding capacity against the bacterial treatment under these conditions and no adhesion or docking of the bacteria is observed (Fig 1A).”

---

## [Decision Letter · Decision Letter 2]

18 Jul 2023

Host-pathogen interaction: Enterobacter cloacae exerts different adhesion and invasion capacities against different host cell types

PONE-D-23-05465R2

Dear Dr. Hensel,

We’re pleased to inform you that your manuscript has been judged scientifically suitable for publication and will be formally accepted for publication once it meets all outstanding technical requirements.

Kind regards,

Kwame Kumi Asare, Ph.D

Academic Editor

PLOS ONE

Additional Editor Comments (optional):

Reviewers' comments:

Reviewer's Responses to Questions

**Comments to the Author**

1. If the authors have adequately addressed your comments raised in a previous round of review and you feel that this manuscript is now acceptable for publication, you may indicate that here to bypass the “Comments to the Author” section, enter your conflict of interest statement in the “Confidential to Editor” section, and submit your "Accept" recommendation.

Reviewer #2: All comments have been addressed

Reviewer #3: All comments have been addressed

2. Is the manuscript technically sound, and do the data support the conclusions?

Reviewer #2: Yes

Reviewer #3: Yes

3. Has the statistical analysis been performed appropriately and rigorously? 

Reviewer #2: Yes

Reviewer #3: Yes

4. Have the authors made all data underlying the findings in their manuscript fully available?

Reviewer #2: Yes

Reviewer #3: Yes

5. Is the manuscript presented in an intelligible fashion and written in standard English?

Reviewer #2: Yes

Reviewer #3: Yes

6. Review Comments to the Author

Reviewer #2: I already have reviewed the previous version of manuscript and given my response related to the questions mentioned above. The authors have clarified the concerns raised in my review.

Reviewer #3: all comments are addressed - please note that the editor should make a decision about the use of supplementary material.

7. PLOS authors have the option to publish the peer review history of their article (what does this mean?). If published, this will include your full peer review and any attached files.

Reviewer #2: **Yes: **Prabhat Nath Jha

Reviewer #3: No

---

## [Editor Report · Acceptance letter]

21 Jul 2023

PONE-D-23-05465R2 

Host-pathogen interaction: Enterobacter *cloacae* exerts different adhesion and invasion capacities against different host cell types. 

Dear Dr. Hensel:

I'm pleased to inform you that your manuscript has been deemed suitable for publication in PLOS ONE. Congratulations! Your manuscript is now with our production department. 

Kind regards, 

on behalf of

Dr. Kwame Kumi Asare 

Academic Editor

PLOS ONE